# Comparison of Change of Direction Speed Performance and Asymmetries between Team-Sport Athletes: Application of Change of Direction Deficit

**DOI:** 10.3390/sports6040174

**Published:** 2018-12-12

**Authors:** Thomas Dos’Santos, Christopher Thomas, Paul Comfort, Paul A. Jones

**Affiliations:** Directorate of Sport, Exercise & Physiotherapy, University of Salford, Salford, Greater Manchester M6 6PU, UK; c.thomas2@edu.salford.ac.uk (C.T.); p.comfort@salford.ac.uk (P.C.); P.A.Jones@salford.ac.uk (P.A.J.)

**Keywords:** performance deficit, imbalance, turning, 505, symmetry, sprint

## Abstract

The purpose of this study was twofold: (1) to examine differences in change of direction (COD) performance and asymmetries between team-sports while considering the effects of sex and sport; (2) to evaluate the relationship between linear speed, COD completion time, and COD deficit. A total of 115 (56 males, 59 females) athletes active in cricket, soccer, netball, and basketball performed the 505 for both left and right limbs and a 10-m sprint test. All team-sports displayed directional dominance (i.e., faster turning performance/shorter COD deficits towards a direction) (*p* ≤ 0.001, *g* = −0.62 to −0.96, −11.0% to −28.4%) with, male cricketers tending to demonstrate the greatest COD deficit asymmetries between directions compared to other team-sports (28.4 ± 26.5%, *g* = 0.19–0.85), while female netballers displayed the lowest asymmetries (11.0 ± 10.1%, *g* = 0.14–0.86). Differences in sprint and COD performance were observed between sexes and sports, with males demonstrating faster 10-m sprint times, and 505 times compared to females of the same sport. Male soccer and male cricketers displayed shorter COD deficits compared to females of the same sport; however, female court athletes demonstrated shorter COD deficits compared to male court athletes. Large significant associations (*ρ* = 0.631–0.643, *p* < 0.001) between 505 time and COD deficit were revealed, while trivial, non-significant associations (*ρ* ≤ −0.094, *p* ≥ 0.320) between COD deficit and 10-m sprint times were observed. In conclusion, male and female team-sport athletes display significant asymmetries and directional dominance during a high approach velocity 180° turning task. Coaches and practitioners are advised to apply the COD deficit for a more isolated measure of COD ability (i.e., not biased towards athletes with superior acceleration and linear speed) and perform COD speed assessments from both directions to establish directional dominance and create a COD symmetry profile.

## 1. Introduction

The ability to accelerate, reverse, or change movement direction and re-accelerate is an important component of multidirectional sport, and is recognised as change of direction (COD) speed [1]. COD speed tasks require no reaction to a stimulus and are generally classified as pre-planned and closed skills, such as running between the bases in softball/baseball or between the wickets in cricket [2,3,4]. However, COD speed has also been defined as “the ability to change initial direction to a predetermined location and space on a field or court” [5], while providing the physiological and mechanical basis underpinning agility [6]; therefore, highlighting its importance for open-skilled sports [5,7]. Nonetheless, regardless of definition, COD speed is a central component of multidirectional sports and, thus, the ability to examine COD ability is of great importance.

Specifically, the ability to turn 180° quickly and proficiently is an important physical quality in multidirectional sports such as soccer, netball, cricket, and basketball [8,9,10,11,12]. For example, time-motion analysis has revealed soccer players perform ~100 turns of 90–180° [8] and are performed when the team is in and out of possession [13], such as transitioning from defence to attack (and vice versa). Frequently, 180° turns are also performed in netball [10], and in cricket the 180° turn is a fundamental movement for batsmen, whereby approximately 40 turns are performed when scoring 100 runs during a match [12]. Given the importance of 180° COD ability in these aforementioned sports, coaches and practitioners are interested in valid and reliable assessments of 180° COD ability to identify strength and weaknesses in their athletes so that informed decisions can be made regarding the future training for that athlete.

A popular assessment to evaluate 180° COD speed is the 505 test [14], whereby the time to complete 5-m entry, 180° turn, and 5-m exit from a 10-m (or yards) flying sprint approach is recorded. Although the 505 test is easy to perform and has subsequently been included in the testing batteries of numerous sports (rugby, basketball, netball, softball, American football) [6,14,15,16,17,18,19], it should be noted that assessments of COD speed based on completion time only are confounded by linear speed [6,16,20,21], with only 31% of the time during the 505 spent actually changing direction [16]. In light of the issues associated with completion time, the COD deficit (athlete’s 10-m or yard sprint time is subtracted from the 505 time) has been developed and proposed to provide a more “isolated measure of COD ability” [6,16,22], without being biased and influenced by an athlete’s acceleration and linear speed qualities [6,16,21,23]. As such, the COD deficit can be easily calculated and often requires minimal effort because 10-m sprint times are typically measured along with 505 performances in testing batteries [22,24].

In multidirectional sport, it would be advantageous to be equally proficient (balanced/symmetrical) and fast at changing direction from both limbs or directions given the unpredictable nature and agility requirements [24]. Researchers have shown that athletes display significantly faster COD speed completion times (~3–10%) from a particular limb or towards a specific direction [25,26,27,28,29]. This asymmetry in completion time or COD deficit and bias towards a limb or direction is known as directional dominance [25,27]. The finding that athletes display directional dominance is unsurprising because due to laterality, humans will preferentially use one side of the body when performing a motor task, typically resulting in more skilful and, therefore, dominant side [30,31]. However, previous research which has shown directional dominance has examined asymmetries in completion time [25,26,27,28,29], which does not provide an isolated measure of COD ability and is biased by an athlete’s acceleration and linear speed [6,16,21,23].

Recently, overcoming the issues of assessing COD performance solely with completion time, the COD deficit has been recently applied to investigate directional dominance in 180° COD speed (505) performance in female youth netballers [24]. Notably, moderate and significantly greater percentage imbalances (−11.9% vs. −2.3%, *g* = 1.03, *p* < 0.0001); thus, asymmetries were demonstrated in COD deficits compared to 505 times between dominant (D) and non-dominant (ND) directions. Moreover, only two of 43 subjects displayed asymmetries greater than 10% based on 505 times; however, when asymmetries were evaluated for COD deficit, a large proportion of subjects were reclassified, with 21 subjects (49%) exhibiting asymmetries greater than 10% [24]. This finding is concerning because failure to inspect asymmetries in COD deficit and solely comparing 505 completion times between directions could lead to misinterpretations of an athlete’s symmetry in COD ability because of the notably lower asymmetries produced.

Despite the recent popularity of the COD deficit [21,23,32,33], to the best of our knowledge, only one study has explored asymmetries in COD deficit during the 505 and this was limited to youth female netballers [24]. Further understanding of COD ability and asymmetries in athletes from different athletic populations is needed to improve a coach’s ability to prescribe and monitor training for their athletes [34]. Therefore, the primary aim of this study was to determine the magnitude of asymmetries in COD performance for team-sports and examine the differences in COD deficit times and asymmetries between female cricket, netball, soccer, and male basketball, soccer, and cricket athletes, while considering the effect of sex and sport. It is important for athletes to have 180° COD ability in these sports, and there is lack of data in these populations. A secondary aim was to evaluate the relationship between linear speed, COD time measured by the 505, and COD deficit. It was hypothesised that directional dominance would be demonstrated by all athletes, with male athletes demonstrating faster performance for all tasks. Additionally, it was hypothesised that soccer and cricket athletes would demonstrate the largest asymmetries. Furthermore, it was hypothesised that a non-significant relationship between linear speed and COD deficit would be observed.

## 2. Materials and Methods

### 2.1. Subjects

Team-sport athletes (n = 115) from soccer, cricket, and court sports (basketball and netball) participated in this study, with Table 1 presenting subject characteristics including age, height, mass, and playing experience. A minimum sample size of 49 was determined from an a priori power analysis using G*Power (Version 3.1, University of Dusseldorf, Germany) [35]. This was based on previously reported [24] effect size of 0.53 for D and ND differences in 505 COD deficits, a power of 0.95, and type 1 error or alpha level of 0.05. Testing took place during the 5th week of preseason for all athletes, having completed a 4-week strength-endurance mesocycle. All subjects avoided strenuous exercise and rested 48 h prior to testing and attended testing in a fed and hydrated state. Written informed consent was provided by all subjects before participation, with parent or guardian consent of all players under the age of 18. Approval for the investigation was provided by the University of Salford Ethics Committee (ethics approval code: HSCR14/129).

### 2.2. Procedures

A cross-sectional analysis, using a mixed, comparative and correlational design was used, following an associative strategy [36]. A within-subjects, comparative design was used to explore the within-subject magnitudes of directional dominance in 505 completion times and COD deficit demonstrated by male and female team-sport athletes. A between-subject, comparative design was used to compare between-subject performance and asymmetries between sexes and sports. A correlational design was used to explore the relationships between 505 times, COD deficits, and linear speed. 

This study investigated between direction asymmetries in COD deficit and completion time, as measured by the 505, over one testing session. On arrival, all subjects had their body mass (Seca Digital Scales, Model 707, Seca, Birmingham, UK), and standing height (Stadiometer; Model 213, Seca, Birmingham, UK) measured to the nearest 0.1 kg and 0.1 cm, respectively. A standardised warm up was performed by all subjects, in line with previous research [27], consisting of dynamic stretches, low-level bilateral and unilateral plyometric drills, light runs, and sprints.

A 505 test was used to assess COD speed performance [14]. The 505 procedures were in line with previously reported methods [6,24,33]; thus, a brief overview is provided. Testing took place on a third-generation artificial rubber crumb surface (Mondo, SportsFlex, 10 mm; Mondo America Inc., Mondo, Summit, NJ, USA) using single beam “Brower photocell timing Gates” (model number BRO001; Brower, Draper, UT, USA) placed at approximate hip height [37]. Three 505 trials from each limb were performed, in a randomised order, with a 2-min rest period between trials. 505 completion times for each trial were recorded to the nearest 0.001 s. COD deficit was calculated using the formula: mean 505 time − mean 10 m sprint time [6,16]. Sprint testing was performed on the same surface as the COD trials with timing gates placed at 0–10m, in line with previously reported procedures [6,33]. Subjects performed three sprint trials, with two minutes’ rest between trials. The mean performance from each of the three sprint and 505 trials were used for further analysis [6,38]. Dominant COD speed performance was classified as the direction (limb) an athlete displayed faster completion times from, whereas ND COD speed performance was classified as the direction (limb) an athlete displayed slower completion times from; in line with previous research [24,25,27]. The asymmetry index for D and ND COD speed performance was calculated by the formula (D − ND/D × 100) [24,25,27].

### 2.3. Statistical Analyses

Mean ± standard deviation (SD) were calculated for all variables. A Shapiro–Wilks test was used to inspect normality. Intraclass correlation coefficients (ICC) (two-way mixed effects, average measures, absolute agreement) and within-subject coefficient of variation (CV) was calculated as SD/mean × 100 for each subject and then averaged across subjects for each respective sport to assess within-session reliability. Minimum acceptable reliability was determined with an ICC > 0.7 and CV < 15% [39,40]. Magnitude of differences between D and ND directions were assessed with paired sample t-tests or Wilcoxon-sign ranked tests for non-parametric variables, with effect sizes calculated using Hedges’ *g* method [41] and interpreted as trivial (<0.19), small (0.20–0.59), moderate (0.60–1.19), large (1.20–1.99), and very large (2.0–4.0) [42]. 

Multiple 2 × 3 (sex × sport) factorial analyses of variance (ANOVAs) were used to determine the effect of sex (male; female) and sport (cricket; court; soccer) on COD performance, asymmetries, and 10-m sprint times. A Bonferonni-corrected pairwise comparison was used to further analyse the effect of sex and sport when a significant main effect or interaction was observed. Partial eta squared effect sizes were calculated for all ANOVAs with the values of 0.01, 0.06, and 0.15 considered as small, medium, and large, respectively, according to Cohen [43]. For non-parametric data, Mann–Whitney U tests were used compare between sexes or sports for pairwise comparisons, while a Kruskal–Wallis test was used to compare multiple groups. For all pairwise comparisons between sexes or sports, Hedges’ *g* effect sizes were calculated to assess the magnitude of differences.

Relationships between sprint, 505 completion time, and COD deficit were evaluated using Spearman’s correlations. The D and ND sides were examined separately and correlations were evaluated using Hopkins’ scale [44] and interpreted as follows: trivial (0.00–0.09), small (0.10–0.29), moderate (0.30–0.49), large (0.50–0.69), very large (0.70–0.89), nearly perfect (0.90–0.99), and perfect (1.00). The criterion for significance was set at *p* ≤ 0.05. All statistical analyses were performed using SPSS (version 25, IBM, New York, NY, USA) and Microsoft Excel (version 2016, Microsoft Corp., Redmond, WA, USA). 

## 3. Results

### 3.1. Reliability Measures

Descriptive statistics and reliability measures for 10-m sprint, 505, and COD deficit times are presented in Table 2. High and acceptable ICCs (ICC = 0.757–0.987) were observed for all variables across all sports. Similarly, low and acceptable levels of variance were observed for sprint and 505 completion times (CV% = 1.1–3.3) across all sports, while slightly greater, yet acceptable levels of variance were demonstrated for COD deficit times across all sports and directions (CV% = 4.9–12.7), excluding male basketball right COD deficit which displayed high and unacceptable levels of variance (CV% = 15.5).

### 3.2. Comparions in 505 Times and Change of Direction (COD) Deficit between Dominant (D) and Non-Dominant (ND) Directions

Comparisons between D and ND directions for 505 and COD deficit times for all sports are presented in Table 3. Small to moderate significant differences between D and ND directions for 505 times were observed across all sports (*p* ≤ 0.001, *g* = −0.51 to −0.74). All sports displayed moderate significant differences in COD deficits between D and ND directions (*p* ≤ 0.001, *g* = −0.62 to −0.96). Moreover, substantially greater asymmetries were observed for COD deficits compared to 505 times (−18.9 ± 18.7 vs. −3.5 ± 2.9%, *p* < 0.001, *g* = 1.15).

### 3.3. Sex and Sport Comparisons in 10-m Sprint Times, 505 Times, COD Deficits, and COD Asymmetries

Pairwise comparisons for COD performance, asymmetries, and 10-m sprint times between sexes and sports are presented in Appendix A. Figure 1, Figure 2 and Figure 3 illustrate the individual COD deficit and asymmetries between sports.

∙Large and significant main effects for sex (*p* < 0.001, η^2^ = 0.440, power = 1.000) and sport (*p* < 0.001, η^2^ = 0.213, power = 1.000) were found for 10-m sprint times, while no significant interaction effect of sex and sport (*p* = 0.116, η^2^ = 0.039, power = 0.440) for 10-m sprints times was revealed. On average, male athletes (*p* < 0.001, *g* = −1.48) and court-sport athletes demonstrated the fastest 10-m sprint times (*p* ≤ 0.014, *g* = −0.48 to −0.89), while males were significantly faster than females of the same sport (*p* ≤ 0.005, *g* = −1.62 to −1.81) (Table 2 and Appendix A).∙A large and significant main effect for sex (*p* < 0.001, η^2^ = 0.322, power = 1.000) was found for D 505 times, but no significant main effect for sport (*p* = 0.211, η^2^ = 0.028, power = 0.328) was observed. A medium and significant interaction effect of sex and sport (*p* = 0.007, η^2^ = 0.088, power = 0.824) for D 505 times was observed. On average, male athletes (*p* < 0.001, *g* = −1.28) and court-sport athletes demonstrated the fastest D 505 times (*p* = 0.325–0.457, *g* = −0.25 to −0.29) (Table 3 and Appendix A), while males displayed shorter D 505 times compared to females of the same sport (*g* = −0.67 to −1.64) (Table 3 and Appendix A).●A large and significant main effect for sex (*p* < 0.001, η^2^ = 0.251, power = 1.000) was found for ND 505 times, but no significant main effect for sport (*p* = 0.211, η^2^ = 0.028, power = 0.328) was observed. A medium and significant interaction effect of sex and sport (*p* = 0.001, η^2^ = 0.115, power = 0.925) for ND 505 times was observed. On average, male athletes (*p* < 0.001, *g* = −1.02) and court-sport athletes demonstrated the fastest ND 505 times (*p* = 0.177–0.193, *g* = −0.32 to −0.41) (Table 3 and Appendix A). While male soccer and male cricketers displayed significantly faster ND 505 times (*p* < 0.001, *g* = −1.45 to −1.53) compared to females of the same sport (Table 3 and Appendix A), a non-significant and trivial difference was observed between sexes for court sports (*p* = 1.000, *g* = 0.17) (Table 3 and Appendix A).●A medium and significant main effect for sport (*p* = 0.007, η^2^ = 0.087, power = 0.820) was found for D COD deficits, but no significant main effect for sex (*p* = 0.362, η^2^ = 0.008, power = 0.148) or interaction effect of sex and sport (*p* = 0.051, η^2^ = 0.053, power = 0.580) for D COD deficits was observed. Overall, males displayed slightly smaller D COD deficits (*p* = 0.261, *g* = 0.21) compared to females, and soccer athletes displayed significantly shorter D COD deficits than other cricket and court-sports (*p* ≤ 0.027, *g* = −0.60 to −0.74) (Table 3 and Appendix A). On average, male soccer and male cricketers tended to display smaller D COD deficits compared to females of the same sport (*g* = −0.34 to −0.58); however, female court athletes displayed smaller D COD deficits compared to male court athletes (*g* = −0.44) (Table 3 and Appendix A).●A small and significant main effect for sport (*p* = 0.049, η^2^ = 0.054, power = 0.588) was found for ND COD deficits, but no significant main effect for sex (*p* = 0.941, η^2^ = 0.000, power = 0.051) was observed. A medium and significant interaction effect of sex and sport (*p* = 0.013, η^2^ = 0.077, power = 0.763) for ND COD deficits was observed. Soccer athletes displayed smaller ND COD deficits than cricket and court-sports (*p* ≤ 0.078, *g* = −0.43 to −0.52) (Table 3 and Appendix A), while on average, male soccer and male cricketers tended to display smaller ND COD deficits compared to females of the same sport (*g* = −0.22 to −0.49). However, female court athletes displayed shorter ND COD deficits (*g* = −0.75) compared to male court athletes (Table 3 and Appendix A).●As COD deficit and 505 imbalance data were non-parametric, the 2 × 3 factorial ANOVA could not be performed. On average, males displayed greater asymmetries than females for COD deficit (*p* = 0.051, *g* = −0.45) and 505 (*p* = 0.026, *g* = −0.47) tasks (Table 3 and Appendix A). Kruskal–Wallis tests revealed no significant differences in COD deficit or 505 asymmetries between sports (*p* = 0.067–0.166), though court-sport athletes tended to display the lowest asymmetries on average (*g* = 0.21–0.35) (Table 3 and Appendix A).

### 3.4. Relationships between 505 Times, COD Deficit, and 10-m Sprint Times

Spearman’s correlation data are presented for D and ND COD performance in Table 4 and scatter plots are presented in Appendix A. Large significant positive associations (*ρ* = 0.631–0.643, *p* < 0.001) between 505 time and COD deficit were observed for both D and ND COD performance. Large significant positive associations (*ρ* = 0.656–0.662, *p* < 0.001) between 505 time and 10-m sprint times were observed for both D and ND COD performance. Trivial, non-significant associations (*ρ* ≤ −0.094, *p* ≥ 0.320) were revealed between COD deficit and 10-m sprint times for both D and ND directions.

## 4. Discussion

The primary aim of this study was to determine the magnitude of asymmetries in COD performance in three male (basketball, soccer, and cricket) and three female (cricket, netball, soccer) team-sports and compare COD deficit and 505 times, asymmetries in COD deficit and 505 times, and 10-m sprints between team-sports, while considering the effect of sex and sport. The main finding was all team-sports displayed directional dominance (Table 2), demonstrating significant asymmetries in 505 and COD deficit times between D and ND directions (Table 2), supporting the study hypotheses. Additionally, in general, sex and sport differences in sprint and COD performance were observed, with males demonstrating faster 10-m sprint times and 505 times compared to females of the same sport (Table 3 and Appendix A). However, while male soccer and male cricketers displayed shorter COD deficits compared to females of the same sport, female court athletes demonstrated shorter COD deficits compared to male court athletes (Table 3 and Appendix A). On average, male cricketers tended to display the greatest asymmetries in COD performance, while female netballers displayed the lowest asymmetries in COD performance (Table 2 and Table 3). Furthermore, a secondary aim was to evaluate the relationship between linear speed, COD time measured by the 505, and COD deficit. Interestingly, supporting the study hypotheses, trivial, non-significant associations were revealed between COD deficit and 10-m sprint times for both D and ND directions (Table 4), indicating that COD deficit provides an isolated measure of COD ability and is not biased towards athletes with superior acceleration and linear speed.

Substantiating the findings of previous research [25,26,27,28,29], all male and female team-sports demonstrated directional dominance (Table 3), displaying superior 505 performance towards the D direction. However, as COD deficit provides a more isolated measure of COD ability [6,16,22], it would be more appropriate to compare asymmetries in COD deficit compared to completion times. Consequently, supporting the findings of Dos’Santos et al. [24], substantially greater asymmetries were observed for COD deficit compared to 505 completion times, with all team-sports demonstrating moderate significant asymmetries in COD deficits (~11–28%) between directions (Table 3, Figure 2). Dos’Santos et al. [24] is the only other study to directly compare COD deficits between D and ND directions reporting significant asymmetries in youth female netball athletes (11.9 ± 12.8%, *p* < 0.001, *g* = −0.53), and although outside of the scope their study, an −8.6% asymmetry to the D side would have been observed in male cricketers in the study by Nimphius et al. [6]. As such, it is clear that team-sport athletes display superior COD performance towards a particular direction (Figure 2 and Figure 3); therefore, practitioners are recommended to evaluate asymmetries in COD performance based on COD deficits for improved profiling regarding an athlete’s symmetry in COD ability [24]. Performing such an analysis will provide further insight into an athlete’s strengths and weaknesses, so that informed training interventions can be designed to correct such deficiencies.

In multidirectional sport, it would be desirable and advantageous for athletes to be equally proficient and fast changing direction from either limb or direction. In this study, males generally demonstrated greater asymmetries, with male cricketers, on average, demonstrating the greatest COD deficit asymmetries compared to other sports (Table 3 and Appendix A, Figure 2), followed by: female soccer, male basketball, male soccer, female cricket, and female netball teams, respectively. As stated earlier, directional dominance has been observed by researchers [24,25,26,27,28,29]. The finding from the present study that athletes display directional dominance is unsurprising because due to laterality, humans will preferentially use one side of the body when performing a motor task, typically resulting in a more skilful and, therefore, dominant side [30,31]. Specifically, male cricketers and male and female soccer players displayed the greatest asymmetries in COD deficits between directions compared to the lower, yet significant asymmetries observed in female netball athletes (Table 2 and Table 3, Figure 2). 

In theory, the greater asymmetries in COD ability observed for male cricketers and soccer athletes could be explained by the asymmetrical nature of theses sports. For example, in cricket, repeated asymmetrical actions of batting, bowling, and turning between the wickets are performed, while in soccer, athletes typically perform repeated kicking actions with a preferred limb which could also lead to the development of asymmetries [30,45,46,47,48,49,50,51]. Conversely, netball is arguably a more symmetrical sport with less demand on lower body asymmetrical actions such as the kicking, batting, bowling actions in soccer and cricket. Nevertheless, it is important to acknowledge that the aforementioned comparisons across sports were based on the means; thus, the individual variation of the COD deficit asymmetries must be considered because athletes from each sport display low and high imbalances (Figure 2 and Figure 3).

Figure 2 and Figure 3 illustrate the individual asymmetries in COD deficits for all team-sports and from 115 athletes, 68, 51, 38, 21, and 9 athletes displayed COD deficit asymmetries greater than 10%, 15%, 20%, 30%, and 50%, respectively. Moreover, Figure 3 illustrates 76 of 115 (66%) athletes displayed faster performance from their right limb. These findings show that team-sport athletes display performance deficits when turning 180° from a particular limb/towards a particular a direction. It was beyond the scope of this study to identify the biomechanical (kinetic and kinematic) mechanisms which explain differences in COD ability between directions; however, preliminary evidence suggests that differences in braking strategies may partially explain asymmetries in 180° turning performance [22,52]. For example, Thomas et al. [52] reported differences in braking strategies during a modified 505 in female soccer players when comparing turns from the preferred and non-preferred kicking limbs. Female soccer players displayed greater magnitudes of penultimate foot contact horizontal braking forces (i.e., braking with the ND limb) when changing direction from the D limb, but this was not the case when changing direction from the ND limb, whereby greater emphasis was placed on the final foot contact. Faster performance was also found for the turns from the kicking limb which could partially be attributed to the female athletes displaying greater horizontal braking forces in the penultimate foot contact relative to the final foot contact; thus, a greater horizontal ground reaction force ratio in their D limb turn. This finding is noteworthy because high magnitudes of penultimate foot contact braking forces relative to the final foot contact have been identified as a determinant of faster 180° turning performance [53,54]. Furthermore, Nimphius et al. [22] reported a 27.4% asymmetry in COD deficit for a female athlete, and attributed the differences based on qualitative assessments of the braking and turning strategies between directions. Joint–joint coordination differences have also been hypothesised as an explanation of asymmetries in COD ability [24], though further research is necessary to substantiate this claim. In order to improve our understanding of asymmetries in turning performance, further research is necessary inspecting the biomechanical and coordination differences between directions.

To the best of our knowledge, this is the first study to compare COD deficit and 505 performance across a range of team-sports, while considering the effect of sex and sport. The COD deficits presented for the team-sports in Table 1 and Table 2 are in line with those reported in male cricket (D: 0.617 ± 0073 s, ND: 0.670 ± 0.087 s) [6], female rugby (left: 0.581 ± 0.248 s, right: 0.545 ± 0.244 s) [23], and Division 1 (0.49 ± 0.16 s) and 2 (0.72 ± 0.18 s) female soccer players [32] using the same assessment (10-m 505). In most cases, sex or sport differences were observed for COD performance and 10-m sprint times (Table 2, Table 3 and Appendix A), with males displaying faster 10-m sprint times compared to females of the same sport (Appendix A). Additionally, medium and significant interaction effects of sex and sport were observed for D and ND 505 times, with males demonstrating faster 505 times compared to females of the same sport (Table 3 and Appendix A). Interestingly, however, male soccer and male cricketers displayed shorter COD deficits compared to females of the same sport, whereas female court athletes demonstrated shorter COD deficits compared to male court athletes (Table 3 and Appendix A), Nonetheless, These findings are consistent with previous research that found males to display faster COD speed performance or exit velocity compared to females [55,56,57]. It is important to note, though, that these comparisons were based on group means, and Figure 1 illustrates that individual variation exists between sports and that some female athletes can display superior COD performance compared to males.

Lower-limb strength and rapid force production qualities have been identified as physical qualities associated with COD speed performance [19,29,33,58,59], which is unsurprising because braking and propulsive forces are associated with faster COD speed performance [53,60,61]. Spiteri et al. [56] compared offensive agility between sexes and found male athletes demonstrated greater isometric strength, greater braking and propulsive forces and impulses (*p* = 0.010, Effect size (ES) = 1.19–2.28), and subsequently greater post stride velocity (*p* = 0.001, ES = 0.75–0.83), although this was in response to a human stimulus at a controlled approach velocity. Schreurs et al. [57] found male athletes displayed faster COD completion times (45°, 90°, 135°, and 180°) compared to females which could be partially attributed to the greater approach velocities demonstrated by the male athletes; a key determinant of faster COD performance [62,63]. Moreover, Condello et al. [64] found male athletes displayed greater vertical (*p* = 0.051, ES = 0.67) and medio-lateral (*p* = 0.005, ES = 1.05) ground reaction forces compared to female athletes during a 60° cut. This finding is noteworthy because greater ground reaction forces are associated with faster performance [53,60,61]; however, Condello et al. [64] failed to compare completion times. Unfortunately, to the best of our knowledge, no other biomechanical research has comprehensively compared COD biomechanics between sexes from a performance perspective, with most studies focusing on injury risk biomechanics between sexes [65,66,67,68,69,70,71,72]. Nevertheless, although strength characteristics and COD biomechanics were not examined in this study, males, in general, may display greater strength and rapid force production characteristics [56,73,74], thus enabling them to apply greater braking and propulsive forces, which may explain the generally superior 505 and COD deficit performances compared to female sports observed in this study. Further research is required comparing strength and biomechanical differences between sexes in relation to COD speed performance.

Consistent with the results of previous studies [6,16,21,23,24], large significant relationships between 505 time and COD deficit (Table 4), were observed for both D and ND directions, indicating that athletes with shorter COD deficits produced faster 505 times. Importantly, however, substantiating the results of previous research [6,16,21,23], trivial, non-significant associations (Table 4) were revealed between COD deficit and 10-m sprint times for both D and ND directions, while a large significant positive association (Table 4) between 505 time and 10-m sprint times were observed for both D and ND COD performance. The results of this study corroborate the findings from previous studies [6,16,20,21,23] that found COD speed assessments based on completion time are biased towards athletes with superior acceleration and linear speed capabilities, whereas assessment-based COD deficits are not biased towards acceleration and linear speed and thus, provide a more isolated measure of COD ability. 

It is worth acknowledging that this study only compared male and female soccer, male and female cricket, male basketball, and female netball team-sports; thus, caution is advised generalising these findings to different athletic populations. As such, further research is needed exploring COD deficits in multidirectional athletes from different athletic populations. Additionally, this study investigated the traditional 505 which only reflects high-entry velocity 180° COD ability. As the biomechanical demands of COD are angle- and velocity-dependent [7,22,75], inspection of COD deficit asymmetries during different angled tests is required (i.e., 45°, 90° and 135°). Moreover, it is worth acknowledging that the testing took place on a standardised surface to permit comparisons across sports, but these may not reflect the surfaces performed in their sports. Furthermore, athletes, such as cricketers, may perform 180° turns holding a bat or implement, although it is worth noting that very strong relationships between 505 completion times and time to run a quick single, run a two, and run a three while holding a cricket bat (*r* = 0.804–0.934, *p* < 0.01) have been observed [76], highlighting the shared variance between the 505 and running between the wickets in cricket. Finally, as stated earlier, although beyond the scope of this study, qualitative video analysis, three-dimensional motion kinetic or kinematic analysis, or muscle strength asymmetry assessments to identify the causes of asymmetry in COD ability were not performed and are, therefore, recommended areas of future research.

## 5. Conclusions

In conclusion, the current study indicates that male and female team-sport athletes display significant asymmetries and directional dominance during a high approach velocity 180° COD task, with substantially greater asymmetries observed for COD deficits compared to completion times. From 115 athletes, 68, 51, 38, 21, and 9 athletes displayed COD deficit asymmetries greater than 10%, 15%, 20%, 30%, and 50%, respectively, and, on average, male cricketers tended to display the greatest asymmetries in COD performance, while female netballers displayed the lowest asymmetries in COD performance. Although it is unclear the causes of directional dominance and why the magnitudes of asymmetry differ between sports, the results from this study show that team-sport athletes display a significant performance deficit when turning direction from the ND limb. 

Additionally, trivial, non-significant associations were revealed between COD deficit and 10-m sprint times for both D and ND directions, indicating that COD deficit provides an isolated measure of COD ability and is not biased towards athletes with superior acceleration and linear speed. As such, the results from the study provide normative COD speed and asymmetry data in male and female team-sport athletes which coaches can use to compare and monitor their findings to. Coaches and practitioners are advised to apply the COD deficit for a more isolated measure of COD ability and perform COD speed assessments from both directions to establish directional dominance and create a COD symmetry profile. 

## Figures and Tables

**Figure 1 sports-06-00174-f001:**
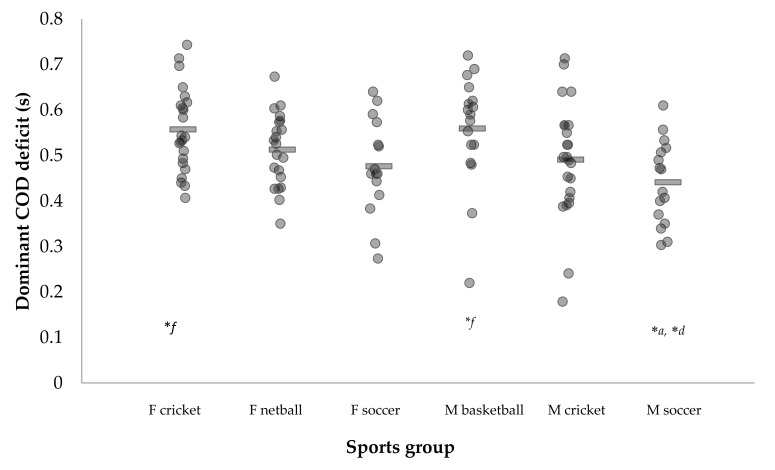
Individual dominant change of direction deficit times across sports. Note: rectangle represents group mean. F: Female; M: Male; **a*: Significantly different to F cricket (*p* < 0.05); **d*: Significantly different to M basketball (*p* < 0.05); **f*: Significantly different to M soccer (*p* < 0.05).

**Figure 2 sports-06-00174-f002:**
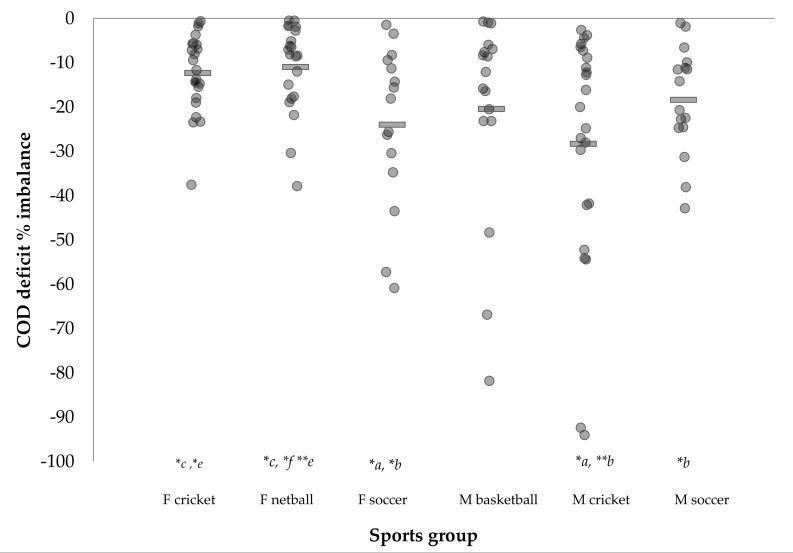
Individual change of direction deficit imbalances across sports. Note: rectangle represents group mean. F: Female; M: Male; **a*: Significantly different to F cricket (*p* < 0.05); **b*: Significantly different to F netball (*p* < 0.05); **c*: Significantly different to F soccer (*p* < 0.05); **d*: Significantly different to M basketball (*p* < 0.05); **e*: Significantly different to M cricket (*p* < 0.05); **f*: Significantly different to M soccer (*p* < 0.05); ** (*p* < 0.01).

**Figure 3 sports-06-00174-f003:**
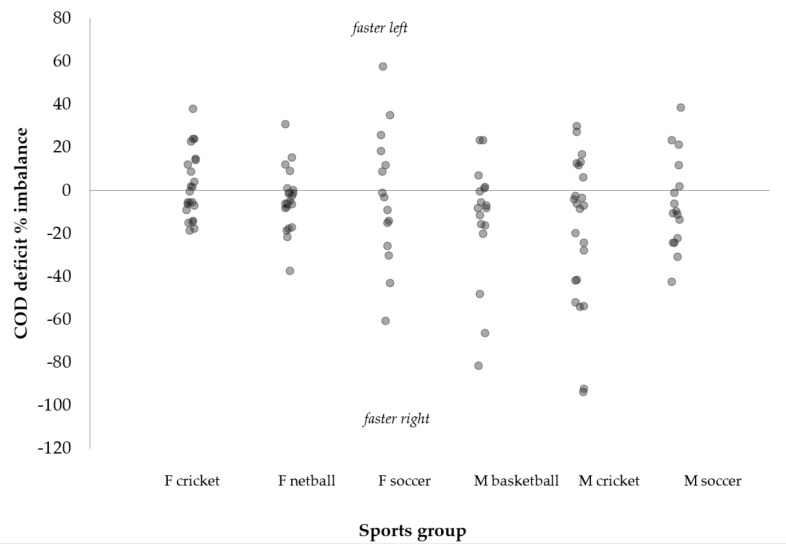
Individual change of direction deficit imbalances across sports showing right or left directional dominance.

**Table 1 sports-06-00174-t001:** Subject characteristics by sport.

Sport	n	Age (years)	Height (cm)	Mass (kg)	Playing Experience (years)
Male Basketball	17	17.3 ± 0.6	187.1 ± 9.4	81.6 ± 10.5	6.2 ± 1.2
Male Cricket	23	18.7 ± 2.7	175.8 ± 6.1	76.9 ± 13.3	6.7 ± 1.7
Male Soccer	16	20.1 ± 0.6	179.1 ± 5.2	76.0 ± 8.6	7.2 ± 1.3
Female Netball	21	18.1 ± 1.1	174.0 ± 6.1	66.7 ± 5.1	6.8 ± 2.0
Female Cricket	23	17.6 ± 1.6	165.2 ± 9.2	61.5 ± 11.1	6.1 ± 1.5
Female Soccer	15	20.6 ± 0.6	168.0 ± 7.2	56.2 ± 6.3	7.0 ± 1.6

**Table 2 sports-06-00174-t002:** Descriptive statistics and reliability measures for 10-m sprint and change of direction speed performance.

Variable	Sport	Mean	SD	ICC	95% LB	95% UB	CV%	95% LB	95% UB
10-m sprint (s) ^†‡^	Female cricket	2.059	0.099	0.935	0.869	0.970	1.9	1.5	2.4
	Female netball	1.968	0.063	0.938	0.872	0.973	1.1	0.8	1.5
	Female soccer	2.139	0.141	0.979	0.950	0.992	1.4	0.9	1.9
	Male basketball	1.854	0.074	0.775	0.503	0.911	2.3	1.1	3.4
	Male cricket	1.884	0.091	0.933	0.867	0.969	1.6	1.0	2.2
	Male soccer	1.932	0.092	0.935	0.852	0.975	1.7	1.1	2.2
505 left (s)	Female cricket	2.646	0.134	0.935	0.869	0.970	1.9	1.5	2.4
	Female netball	2.517	0.081	0.953	0.896	0.980	1.1	0.8	1.3
	Female soccer	2.672	0.197	0.931	0.835	0.975	2.4	1.4	3.5
	Male basketball	2.492	0.158	0.906	0.791	0.963	2.9	2.2	3.7
	Male cricket	2.458	0.133	0.899	0.800	0.954	2.4	1.8	3.1
	Male soccer	2.425	0.120	0.927	0.833	0.972	2.0	1.4	2.6
505 right (s)	Female cricket	2.652	0.125	0.932	0.864	0.969	1.9	1.5	2.3
	Female netball	2.499	0.098	0.889	0.772	0.952	1.3	0.6	2.1
	Female soccer	2.668	0.235	0.987	0.970	0.995	1.3	0.8	1.8
	Male basketball	2.433	0.130	0.783	0.512	0.915	3.3	1.9	4.6
	Male cricket	2.401	0.173	0.900	0.802	0.954	2.7	1.6	3.8
	Male soccer	2.401	0.135	0.919	0.816	0.969	2.4	1.8	3.1
COD deficit left (s)	Female cricket	0.587	0.107	0.897	0.793	0.953	8.6	6.8	10.5
	Female netball	0.548	0.071	0.939	0.865	0.974	4.9	3.9	5.9
	Female soccer	0.533	0.106	0.757	0.414	0.912	12.7	6.9	18.5
	Male basketball	0.638	0.137	0.876	0.725	0.951	11.6	8.8	14.5
	Male cricket	0.575	0.094	0.799	0.599	0.908	10.6	7.7	13.5
	Male soccer	0.493	0.097	0.888	0.742	0.957	10.1	7.1	13.1
COD deficit right (s)	Female cricket	0.593	0.097	0.886	0.772	0.948	8.6	6.6	10.5
	Female netball	0.530	0.105	0.903	0.800	0.957	6.0	3.3	8.7
	Female soccer	0.529	0.170	0.976	0.943	0.991	6.6	4.3	8.9
	Male basketball	0.579	0.143	0.820	0.595	0.929	15.5	7.5	23.4
	Male cricket	0.517	0.148	0.864	0.730	0.937	11.3	7.3	15.3
	Male soccer	0.469	0.117	0.892	0.753	0.959	12.5	9.0	16.0

Key: COD: Change of direction; ICC: Intraclass correlation coefficient; CV%: Coefficient of variation; LB: Lower bound confidence interval; UB: Upper bound confidence interval. ^†^: Significant main effect for sport (*p* < 0.05); ^‡^: Significant main effect for sex (*p* < 0.05).

**Table 3 sports-06-00174-t003:** Dominant versus non-dominant comparisons for 505 and COD deficit times.

Sport	505 (s)								COD Deficit (s)							
	D ^‡≠^		ND ^‡≠^		Imbalance (%)		*p*	*g*	D ^†^		ND ^†≠^		Imbalance (%)		*p*	*g*
	Mean	SD	Mean	SD	Mean	SD			Mean	SD	Mean	SD	Mean	SD		
F cricket	2.616	0.125	2.682	0.125	−2.6	1.8	<0.001	−0.52	0.557	0.093	0.623	0.099	−12.4	8.9	<0.001	−0.68
F netball	2.481	0.075	2.535	0.096	−2.2	2.0	<0.001	−0.62	0.512	0.081	0.566	0.090	−11.0	10.1	<0.001	−0.62
F soccer	2.615	0.198	2.724	0.220	−4.2	3.4	0.001	−0.51	0.476	0.106	0.585	0.149	−24.0	18.4	0.001	−0.82
M basketball	2.413	0.118	2.513	0.157	−4.1	4.0	0.001	−0.70	0.559	0.123	0.658	0.144	−20.5	23.5	0.001	−0.73
M cricket	2.374	0.163	2.485	0.128	−4.8	3.3	<0.001	−0.74	0.490	0.129	0.601	0.097	−28.4	26.5	<0.001	−0.96
M soccer	2.373	0.108	2.453	0.134	−3.3	2.0	<0.001	−0.64	0.441	0.092	0.520	0.107	−18.5	12.2	<0.001	−0.78

Key: D: Dominant; ND: Non-dominant; COD: Change of direction; F: Female; M: Male. ^†^: Significant main effect for sport (*p* < 0.05); ^‡^: Significant main effect for sex (*p* < 0.05); ^≠^: Significant interaction effect of sport and sex (*p* < 0.05).

**Table 4 sports-06-00174-t004:** Spearman’s correlations between 505 time, COD deficit, and 10-m sprint time for D and ND directions (pooled data, n = 115).

Correlation	D 505 vs. D COD Deficit	D 505 vs. 10-m Sprint	D COD Deficit vs. 10-m Sprint
*ρ*	0.643	0.656	−0.087
*p* value	<0.001	<0.001	0.353
**Correlation**	**ND 505 vs. ND COD Deficit**	**ND 505 vs. 10-m Sprint**	**ND COD Deficit vs. 10-m Sprint**
*ρ*	0.631	0.662	−0.094
*p* value	<0.001	<0.001	0.320

Key: D: Dominant; ND: Non-dominant; COD: Change of direction.

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
