# Peer review of "Comparison of Change of Direction Speed Performance and Asymmetries between Team-Sport Athletes: Application of Change of Direction Deficit"

_sports, 2018, doi:10.3390/sports6040174_

Round 1
Reviewer 1 Report
The purpose of study was examine differences in COD and asymmetries between team sports and evaluate the relationship between linear speed, COD and COD déficit.
The objective is interesting and relevant. The sample is adequate, but there are different aspects related to the method and the analysis model, as well as the results interpretation that should be better addressed.
1. The reason of the need and relevance of the study is well developed.
2. The results are presented according to gender, so the objectives (line 88) should already be formulated in this sense.
3. Line 99. Years of experience in sports practice when it comes to young athletes should be part of the characteristics of the sample.
4. Line 118. The research design must be more developed. From my knowledge you do not used a unique design, but there are at least two: a correlational one and another comparative following an associative strategy.
5. Line 138. Why did you not choose the best time of the three attempts for the analysis? Would not it be more appropriate in terms of sports performance?
6. Line 142. Were all participants measured in the same conditions? That is, what day within the training microcycle was chosen to perform the tests. It is important to contextualize that the assesments were made at a specific point of the season since this can condition the results.
7. Line 144. The choice of the dominant side should be better explained.
8. Line 151. Looking at the results of the CV, these do not match the formula applied. Hopkins proposes a logarithmic function for its calculation that would be applicable to this case. Please check this because the results could be modified substantially.
9. Line 153. The statistical analysis is very "compartmentalized" which makes it lose statistical power. Perhaps the use of a mixed ANOVA with post hoc and with the eta partial squared calculation would allow a greater solidity in the results. However for this, all variables should be normal what I understand that it might be a "handicap". Could it be possible to do it?
10. There is no statistical treatment for the variable "gender", however the results are shown in terms of this variable, so I suggest that this point be explained.
11. Line 174, from my point of view the results of the ICC must be expressed according to the Confidance Interval (95%). To continue with the editorial line you can consult the work “Intrasession Reliability of the Tests to Determine Lateral Asymmetry and Performance in Volleyball Players” Symmetry 2018, 10, 416; doi:10.3390/sym10090416.
12. The discussion has not been deeply reviewed since the possible changes in the results could modify it to some extent. I will wait for your news to do later the revision of that section.
Author Response
Reviewer 1:
The purpose of study was examine differences in COD and asymmetries between team sports and evaluate the relationship between linear speed, COD and COD déficit.
The objective is interesting and relevant. The sample is adequate, but there are different aspects related to the method and the analysis model, as well as the results interpretation that should be better addressed.
1. The reason of the need and relevance of the study is well developed.
Response: Thank you for your comment.
2. The results are presented according to gender, so the objectives (line 88) should already be formulated in this sense.
Response: Thank you for your comment. In light of your suggestions, to consider the effect of sex and sport, we have now performed a 2 x 3 factorial ANOVA (sex x sport) on parametric data 10-m sprint times, 505 times, and COD deficit. Consequently, the results section 3.3 has been completely amended, and some parts of the abstract and discussion have been amended also. We have now stated in our aims that we are considering the effect of sex and sport.
3. Line 99. Years of experience in sports practice when it comes to young athletes should be part of the characteristics of the sample.
Response: Thank you for your comment. We have included subject characteristics into a Table, and included playing experience (years) (see Table 1).
4. Line 118. The research design must be more developed. From my knowledge you do not used a unique design, but there are at least two: a correlational one and another comparative following an associative strategy.
Response: Thank you for your comment. We have amended the procedure sections, outlining the research designs in more detail.
It is has been amended to:
“A cross-sectional analysis, using a mixed, comparative and correlational design was used, following an associative strategy (Ato et al., 2013). A within-subjects comparative design was used to explore the within-subject magnitudes of directional dominance in 505 completion times and COD deficit demonstrated by male and female team-sport athletes. A between-subject comparative design was used to compare between-subject performance and asymmetries between sports. A correlational design was used to explore the relationships between 505 times, COD deficits, and linear speed.”
This is in line with Ato et al. (2013) recommendations.
5. Line 138. Why did you not choose the best time of the three attempts for the analysis? Would not it be more appropriate in terms of sports performance?
Response: Thank you for your comment. As we were primarily looking at COD deficit, Nimphius et al. (2013;2016) states to use mean 505 time – mean 10 m time for the COD deficit calculation. In addition, Al Haddad et al. (2015) showed best or average approaches show minimal differences and very similar outcomes.
Al Haddad, H., Simpson, B. M., & Buchheit, M. (2015). Monitoring changes in jump and sprint performance: best or average values?. International journal of sports physiology and performance, 10(7), 931-934.
6. Line 142. Were all participants measured in the same conditions? That is, what day within the training microcycle was chosen to perform the tests. It is important to contextualize that the assessments were made at a specific point of the season since this can condition the results.
Response: Thank you for your comment. Yes, we state that the athletes were tested in the preseason phase, and we have now stated that they were in the 5th week of preseason and that they had just completed a 4-week strength-endurance mesocycle. We have also stated they rested for 48 hours the day before testing.
7. Line 144. The choice of the dominant side should be better explained.
Thank you for your comment. In the introduction we state that an asymmetry in completion time or COD deficit and bias towards a limb or direction is known as directional dominance (Hart et al., 2015; Dos’Santos et al., 2017). The 505 was performed in both directions (i.e. turn left and turn right), thus the side that produced the fastest time was considered dominant, and the side with the slowest was considered the slowest. D or ND did not refer to kicking limb.
As such, we have amended the section:
“Dominant COD speed performance was classified as the direction (limb) an athlete displayed faster completion times from, whereas ND COD speed performance was classified as the direction (limb) an athlete displayed slower completion times; in line with previous research [24,25,27].”
8. Line 151. Looking at the results of the CV, these do not match the formula applied. Hopkins proposes a logarithmic function for its calculation that would be applicable to this case. Please check this because the results could be modified substantially.
Response: Thank you for your comment. Apologies, this should have been made clearer. We report the within-subject CV%. The within-subject coefficient of variation (CV) was calculated as SD/mean x 100 for each subject and then averaged across subjects for each respective sport. We have amended the statistical analysis section.
9. Line 153. The statistical analysis is very "compartmentalized" which makes it lose statistical power. Perhaps the use of a mixed ANOVA with post hoc and with the eta partial squared calculation would allow a greater solidity in the results. However for this, all variables should be normal what I understand that it might be a "handicap". Could it be possible to do it?
Response: Thank you for your comment. In light of your suggestions, to consider the effect of sex and sport, we have now performed a 2 x 3 factorial ANOVA (sex x sport) on parametric data 10-m sprint times, 505 times, and COD deficit. Consequently, the results section 3.3 has been completely amended, and some parts of the abstract and discussion have been amended also. We have also included partial eta squared and observed power. Pairwise comparisons are presented in Table S1 supplementary material.
10. There is no statistical treatment for the variable "gender", however the results are shown in terms of this variable, so I suggest that this point be explained.
Response: Thank you for your comment. We have addressed this in the above response.
11. Line 174, from my point of view the results of the ICC must be expressed according to the Confidance Interval (95%). To continue with the editorial line you can consult the work “Intrasession Reliability of the Tests to Determine Lateral Asymmetry and Performance in Volleyball Players” Symmetry 2018, 10, 416; doi:10.3390/sym10090416.
Response: Thank you for your comment. The 95% confidence intervals were already included in Table 1 in the original manuscript. As Table 1 has now changed to Table 2, please see Table 2 where its presents the 95% confidence intervals for the ICC and the CV%.
12. The discussion has not been deeply reviewed since the possible changes in the results could modify it to some extent. I will wait for your news to do later the revision of that section.
Reviewer 2 Report
Generally, I enjoyed reading the manuscript. Congratulations to authors. The introduction is very nice and informative, methods are clearly described, and the results are well presented. My only suggestion is based on Discussion as it is. Specifically, as far as my opinion is concerned, there are too many "reflections" on Results, and you often repeat numerical values from Tables. I see no reason for such an approach. It sounds like you are trying to "justify" your discussion and ideas, while there is no need for this. Mainly, if you feel that you must repeat your results within the discussion section, it simply means that you did not present is properly where it should be done (in Results section). So better stick to Discussion and avoid repeating numerical results.
There is another thing which deserves attention. You tested athletes on "standardized court", and this is OK with regard to standardization of the measurement. However, it puts the question on the applicability of your results in "real sport settings", don't you think? Mainly, athletes perform CODS scenarios in specific sport settings (i.e. surfaces) and therefore results you have obtained are useful for comparison between sports, but of limited applicability in training. It should be mentioned as a study limitation. Also, in some sports, cricket, for example, athletes run and perform CODS while holding the equipment, and therefore you tested them in "definite" non-specific environment. This also should be mentioned
Author Response
Reviewer 2:
Generally, I enjoyed reading the manuscript. Congratulations to authors. The introduction is very nice and informative, methods are clearly described, and the results are well presented. My only suggestion is based on Discussion as it is. Specifically, as far as my opinion is concerned, there are too many "reflections" on Results, and you often repeat numerical values from Tables. I see no reason for such an approach. It sounds like you are trying to "justify" your discussion and ideas, while there is no need for this. Mainly, if you feel that you must repeat your results within the discussion section, it simply means that you did not present is properly where it should be done (in Results section). So better stick to Discussion and avoid repeating numerical results.
Response: Thank you for your comment. We have avoided repeating numerical results that are already included in tables/figures and removed from the text. However, when we refer to the results from previous research, we have kept their values in the text for comparative purposes.
There is another thing which deserves attention. You tested athletes on "standardized court", and this is OK with regard to standardization of the measurement. However, it puts the question on the applicability of your results in "real sport settings", don't you think? Mainly, athletes perform CODS scenarios in specific sport settings (i.e. surfaces) and therefore results you have obtained are useful for comparison between sports, but of limited applicability in training. It should be mentioned as a study limitation. Also, in some sports, cricket, for example, athletes run and perform CODS while holding the equipment, and therefore you tested them in "definite" non-specific environment. This also should be mentioned
Response: Thank you for your comment. We agree with your point regarding the surface being a limitation and have acknowledged this as a limitation in the final paragraph of the discussion. However, we feel this is a strength of our study to permit fair comparisons between sports.
The purpose of the 505 COD speed assessment is to assess an athlete’s physical capacity to change direction during a sharp, high entry velocity 180˚ turn. The test is not designed to fully simulate sports specific scenarios, bur rather evaluate the ability to rapidly decelerate and reaccelerate during a 180˚ turn. Although the test may not be fully specific to some of the surfaces or actions in sports (i.e. cricketers hold a bat, soccer play on grass), we feel the information provides novels insights into the COD deficit symmetry profiles for athletes. If an athlete demonstrates poor performance during a pre-planned COD speed task, then performance will most certainly be poor during a sport specific task. We could not perform the 180˚ turn task with cricketers holding a sports-specific implement because again this would not be standardised between sports. However, research (Foden et al., 2015) has shown that, for example, in cricket, a strong relationship between 505 completion times and time to run a quick single, run a two, and run a three while holding a cricket bat (r = 0.804-0.934, p <0.01), highlighting the shared variance between the 505 and running between the wickets in cricket. We have acknowledged this is in the final paragraph of the discussion.
Foden, M., Astley, S., Comfort, P., McMahon, J. J., Matthews, M. J., & Jones, P. A. (2015). Relationships between speed, change of direction and jump performance with cricket specific speed tests in male academy cricketers. Journal of Trainology, 4(2), 37-42.
Round 2
Reviewer 1 Report
Undoubtedly, the manuscript has been improved
I have only one relevant question.
The conclusions should stick to the findings and not be speculative
Author Response
Thank you for your comment.
We have removed the speculative statements from the conclusion. Please see the revised conclusion, whereby the speculative sentences have been removed. These are indicated in red font and strikethrough.